# The effect of compound kushen injection on cancer cells: Integrated identification of candidate molecular mechanisms

Jian Cui[1], Zhipeng Qu[1], Yuka Harata-Lee[1], Hanyuan Shen[1], Thazin Nwe Aung[1], Wei Wang[2], R. Daniel Kortschak[1], David L. Adelson[1]*

**1** Department of Molecular and Biomedical Science, University of Adelaide, Adelaide, South Australia, Australia, **2** Zhendong Research Institute, Shanxi-Zhendong Pharmaceutical Co Ltd, Beijing, China

* david.adelson@adelaide.edu.au

**Data Availability Statement:** The data discussed in this publication have been deposited in NCBI's Gene Expression Omnibus (Edgar et al., 2002) and are accessible through GEO Series accession

## Abstract

Traditional Chinese Medicine (TCM) preparations are often extracts of single or multiple herbs containing hundreds of compounds, and hence it has been difficult to study their mechanisms of action. Compound Kushen Injection (CKI) is a complex mixture of compounds extracted from two medicinal plants and has been used in Chinese hospitals to treat cancer for over twenty years. To demonstrate that a systematic analysis of molecular changes resulting from complex mixtures of bioactives from TCM can identify a core set of differentially expressed (DE) genes and a reproducible set of candidate pathways. We used *in vitro* cancer models to measure the effect of CKI on cell cycle phases and apoptosis, and correlated those phenotypes with CKI induced changes in gene expression. We treated two cancer cell lines with or without CKI and assessed the resulting phenotypes by employing cell viability and proliferation assays. Based on these results, we carried out high-throughput transcriptome data analysis to identify genes and candidate pathways perturbed by CKI. We integrated these differential gene expression results with previously reported results and carried out validation of selected differentially expressed genes. CKI induced cell-cycle arrest and apoptosis in the cancer cell lines tested. In these cells CKI also altered the expression of 363 core candidate genes associated with cell cycle, apoptosis, DNA replication/repair, and various cancer pathways. Of these, 7 are clinically relevant to cancer diagnosis or therapy, 14 are cell cycle regulators, and most of these 21 candidates are downregulated by CKI. Comparison of our core candidate genes to a database of plant medicinal compounds and their effects on gene expression identified one-to-one, one-to-many and many-to-many regulatory relationships between compounds in CKI and DE genes. By identifying genes and promising candidate pathways associated with CKI treatment based on our transcriptome-based analysis, we have shown that this approach is useful for the systematic analysis of molecular changes resulting from complex mixtures of bioactives.

number GSE124715 (https://www.ncbi.nlm.nih.gov/geo/query/acc.cgi?acc=GSE124715)

**Funding:** WW: Chinese National Project for Standardization of Chinese Materia Medica (ZYBZH-C-JIN-43), the Special International Cooperation Project of Traditional Chinese Medicine (GZYYGJ2017035) DLA: University of Adelaide-Zhendong Australia - China Centre for Molecular Chinese Medicine. Funders had no control the experiments, study design, data collection and analysis, preparation of the manuscript or decision to publish.

**Competing interests:** We wish to draw the attention of the Editor to the following facts which may be considered as potential conflicts of interest and to significant financial contributions to this work. While a generous donation was used to set up the Zhendong Centre by Shanxi Zhendong Pharmaceutical Co Ltd, This does not alter our adherence to PLOS ONE policies on sharing data and materials. Zhendong Pharmaceutical Co Ltd did not determine the research direction for this work or influence the analysis of the data. JC: no competing interests, ZQ: no competing interests, YHL: no competing interests, HS: no competing interests, TNA: no competing interests, WW: is an employee of Zhendong Pharma seconded to Zhendong Centre to learn bioinformatics methods, RDK: no competing interests, DLA: Director of the Zhendong Centre which was set up with a generous donation from the Zhendong Pharmaceutical Co Ltd. Zhendong Pharmaceutical has had no control over these experiments, their design or analysis and have not exercised any editorial control over the manuscript.

## Introduction

The treatments of choice for cancer are often radiotherapy and/or chemotherapy, and while these can be effective, they can cause quite serious side-effects, including death. These side-effects have driven the search for adjuvant therapies both to mitigate side-effects and/or potentiate the effectiveness of existing therapies. Traditional Chinese Medicine (TCM) is one of the adjuvant therapies, particularly in China, but increasingly so in the West. Although clinical trial data describing the effectiveness of TCM are currently limited, TCM remains an attractive option because its potential effectiveness is believed to result from the cumulative effects of multiple compounds on multiple targets [1]. However, because there has not been enough rigorous evidence-based assessment on the efficacy and function of TCM and because of its alternative theoretical system compared to Western medicine, adoption of its plant-derived therapeutics has been questioned.

CKI is an herbal extract from two TCM plants, Kushen (*Sophora flavescens*) and Baituling (*Smilax Glabra*) and contains more than 200 different chemical compounds including alkaloids and flavonoids such as matrine, oxymatrine, and kurarinol that have been reported to have anti-cancer activities [2–5]. Some of these activities have been shown to influence the expression of TP53, BAX, BCL2, and other key genes which are known to be important in cancer cell growth and survival [6–9]. CKI has been approved by the State Food and Drug Administration (SFDA) of China for clinical use since 1995 [2] (State medical license no. Z14021231) and it has been clinically used to treat a variety of cancers including lung cancer, liver cancer, breast cancer, ovarian cancer and colorectal cancer [10].

We have previously characterized the effect of CKI on the transcriptome of MCF-7 breast carcinoma cells. In this report, we extended our analysis to two additional human cancer cell lines (MDA-MB-231, breast carcinoma, and Hep G2, hepatocellular carcinoma). Both cell lines have also been shown to undergo apoptosis in response to compounds found in CKI [4, 5, 7, 11]. Hep G2 is one of the most sensitive cancer cell lines with respect to exposure to CKI [12] and CKI is often used in conjunction with Western chemotherapy drugs for the treatment of liver cancer patients in China. While the specific mechanism of action of CKI is unknown, several recent studies have reported that CKI or its primary compounds alter the regulation/expression of products of oncogenes/tumor suppressor genes, including CTNNB1, TP53, STAT3, and AKT [2, 4, 13–15].

However, these and other reports did not evaluate the entire range of molecular changes from treatment with a multi-component mixture such as CKI [16, 17]. Whilst several research databases and tools for TCM research have been developed [18–20], they are limited by the fact that most of the studies that contribute to the corpus of these databases are from different experimental systems, use single compounds or measure effects based on one or a handful of genes/gene products.

In contrast to previous studies, our strategy was to carry out comprehensive transcriptome profiling and pathway identification from cancer cells treated with CKI. Instead of focusing on specific genes or pathways in order to design experiments, we have linked phenotypic assessment and RNA-seq analysis to CKI treatment. This allows us to present an unbiased, comprehensive analysis of CKI specific responses of biological networks associated with cancer. Our results indicate that different cancer cell lines that undergo apoptosis in response to CKI treatment can exhibit different CKI induced gene expression profiles that nonetheless implicate similar core genes and pathways in multiple cell lines.

The current study presents the effects of CKI on gene expression in cancer cells with an aim to identify candidate pathways and regulatory networks that may be perturbed by CKI *in vivo*. To this end, we primarily use concentrations of CKI higher than used *in vivo* in order to be

able to detect effects in the short time frames available to tissue culture experiments. We also combine our current analysis with previously published data to focus on a shared, much smaller set of candidate genes and pathways.

## Materials and methods

### Cell culture and reagents

CKI (batch number: 20150404, total alkaloid concentration of 25 mg/ml) in 5 ml ampoules was provided by Zhendong Pharmaceutical Co. Ltd. (Beijing, China) Quality control documentation in S1 File report. Chemotherapeutic agent, Fluorouracil (5-FU) was purchased from Sigma-Aldrich (MO, USA). A human breast adenocarcinoma cell line, MDA-MB-231 (ATCC Number HTB-26, Lot number 62235654) [21] and a hepatocellular carcinoma cell line Hep G2 (ATCC Number HB-8065, Lot number 61490318) [22] were purchased from American Type Culture Collection (ATCC, VA, USA). The cells were cultured in Dulbecco's Modified Eagle Medium (Thermo Fisher Scientific, MA, USA) supplemented with 10% fetal bovine serum (Thermo Fisher Scientific). Both cell lines were cultured at 37°C with 5% $CO_2$.

For all *in vitro* assays, $4 \times 10^5$ cells were seeded in 6-well trays and cultured overnight before being treated with either CKI (at 1 mg/ml and 2 mg/ml of total alkaloids) or 5-FU (150 $\mu$g/ml for Hep G2 and 20 $\mu$g/ml for MDA-MB-231) based on the cell viability assay [23]. We previously examined the effect of vehicle control (vc) used to extract CKI from raw materials on cells and didn't observe any statistically significant differences compared to medium only with respect to the cell viability assay [23]. Therefore, cells treated with medium only were used as negative control and labelled as "untreated". After 24 and 48 hours of treatment, cells were harvested and subjected to the downstream experiments.

### Cell cycle and apoptosis assay

The assay was performed as previously described [24]. For each cell line, three operators performed the assay twice, each containing 3 technical replicates in order to ensure reproducibility of the observations. The triplicate technical replicates were averaged and the mean values (n = 6) were used to calculate the reported results. The results were obtained by flow cytometry using either FACScanto or LSRII (BD Biosciences, NJ, US).

### RNA isolation and sequencing

The treated cells were harvested, and the cell pellets were snap frozen with liquid nitrogen and stored at -80°C. Three biological replicates for each group were collected and total RNA was isolated with PureLink™ RNA Mini Kit (Thermo Fisher Scientific) according to the manufacturer's protocol. After quantified using a NanoDrop Spectrophotometer ND-1000 (Thermo Fisher Scientific), the quality of the total RNA was verified on a Bioanalyzer by Cancer Genome Facility (SA, Australia) ensuring all samples had RINs>7.0.

Libraries were prepared according to the TruSeq Stranded mRNA-seq with dual-indexes protocol and the sequencing was performed on the NextSeq500 v2 platform with 75bp paired-end reads by the Ramaciotti Centre for Genomics (NSW, Australia). The fastq files were generated and quality control trimmed through Basespace with FASTQ Generation *v1.0.0*.

The data discussed in this publication have been deposited in NCBI's Gene Expression Omnibus [25] and are accessible through GEO Series accession number GSE124715 (https://www.ncbi.nlm.nih.gov/geo/query/acc.cgi?acc=GSE124715).

## Bioinformatics analysis of RNA sequencing

The clean Hep G2 reads were aligned to the human genome reference (hg38) using STAR v2.5.1 with following parameters: –outFilterMultimapNmax 20 –outFilterMismatchNmax 10 –outSAMtype BAM SortedByCoordinate –outSAMstrandField intronMotif [26]. The clean MDA-MB-231 reads were aligned to reference genome (hg19) using TopHat2 v2.1.1 with following parameters: –read-gap-length 2 –read-edit-dist 2 [27]. Differential expression analysis for reference genes was performed with edgeR and differentially expressed (DE) genes were selected with a False Discovery Rate<0.05 [28].

The DE genes in common for both Hep G2 and MDA-MB-231 cell lines at 24 hours and 48 hours after CKI treatment were selected as "shared" genes. These shared genes were utilized to describe the major anti-cancer functions and principal mechanisms of CKI.

Gene Ontology (GO), and Kyoto Encyclopedia of Gene and Genomes (KEGG) over-representation analyses of both cell lines were carried out using the online database system ConsensusPathDB [29] with the following settings: "Biological process" at third level (for GO); q values (<0.01) were corrected for multiple testing with the system default settings. Disease Ontology (DO) over-representation analyses of both cell lines were performed by using the Bioconductor R package clusterProfiler v3.5.1 [30]. For the functional analyses of shared/core genes, the method was similar to our previous study [24] using ClueGO app 2.2.5 in Cytoscape v3.6.0. We enriched our GO terms in the biological process category at level 3 and KEGG pathways, showing only terms/pathways with $p$ values less than 0.01. Specific over-represented terms/pathways and gene expression status mapping in KEGG pathways were visualized with the R package "Pathview" [31].

## Gene expression-based investigation of bioactive components in CKI

To integrate with previous data from MCF-7 cells [24], all the shared DE genes regulated by CKI identified in all three cell lines using edgeR were mapped to the BATMAN-TCM database [32]. The pharmacophore modeling method [33] was used to generate the interaction network between the key genes and TCM components using R package igraph [34].

## Reverse transcription quantitative polymerase chain reaction (RT-qPCR)

RT-qPCR was performed as previously described [24]. The list of target genes selected for this study and the sequences of all primers are shown in S1 Table.

# Results

## Effect of CKI on the cell cycle and apoptosis

In our previous study, CKI significantly perturbed/suppressed cancer cell target genes/networks. In the current study, we present results that confirm and generalize our previous work. We had previously determined that low concentrations of CKI in our short-term cell assay showed no/little phenotypic effect within 48 hours, and very high doses resulted in excessive cell death at 48 hours precluding the isolation of sufficient RNA for transcriptome analysis [24]. Therefore, in our current study with the two additional cell lines, to ensure consistency, we selected 1 mg/ml and 2 mg/ml total alkaloid equivalent concentrations of CKI for our assays because they generated reproducible and significant phenotypic effects on both cell lines' viability and apoptosis in our cell culture assay [23, 24].

We used flow cytometric analysis of propidium iodide-stained cells to assess the effect of CKI on cell cycle and apoptosis. In Hep G2 cells, CKI treatment resulted in an overall increase in the proportion of cells in G1 phase and a decrease in S phase (Fig 1a and 1b). Similarly, in

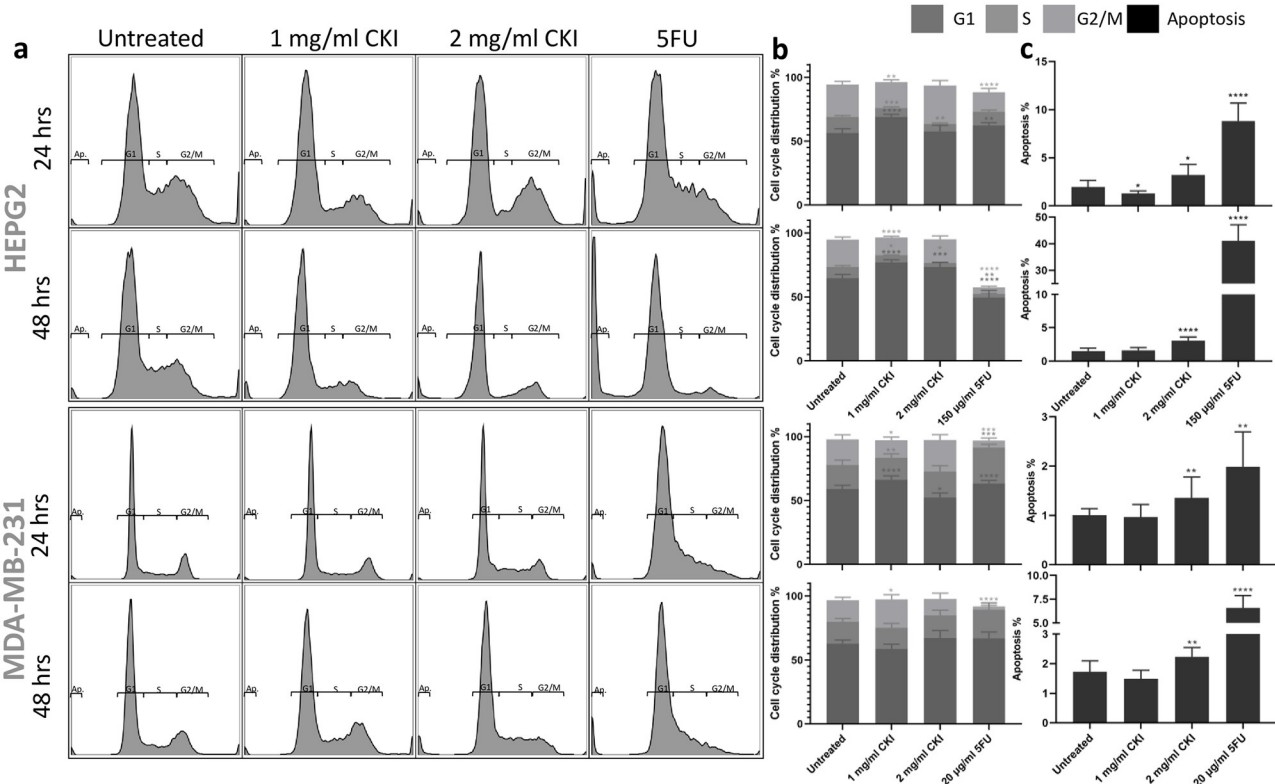

**Fig 1. Effects of different treatment on cell cycle and apoptosis of Hep G2 and MDA-MB-231 cells.** a: The apoptosis and cell cycle distribution of each cell line after 24- and 48-hour treatments with CKI or 5-Fu assessed by PI staining. b: Percentages of cells in different phases of cell cycle resulting from treatment. c: Percentage of apoptotic cells after treatment. Results shown are mean ±95%CI (from 6 triplicates, see Methods). Statistically significant differences from untreated control were identified using two-way ANOVA (*p<0.05, **p<0.01, ****p<0.0001).

MDA-MB-231 cells, although a consistent increase in G1 phase was not observed, CKI caused a decrease in S phase particularly at the 24-hour time point (Fig 1a and 1b) indicating possible cell cycle arrest at G1 phase. Furthermore, CKI consistently induced a significantly higher level of apoptosis in both cell lines at both time points compared to untreated cells at 2 mg/ml concentration (Fig 1c). These data together suggest that CKI has effects on the cell cycle by interfering with the transition between G1 to S phase as well as by acting on the apoptosis pathway and promoting cell death.

## CKI perturbation of gene expression

In order to elucidate the molecular mechanisms of action of CKI on these cancer cells, we carried out transcriptome analysis. As mentioned above, RNA samples from two cell lines were sequenced with 75 bp paired-end reads. We had previously sequenced transcriptomes from CKI treated MCF-7 cells [24] and have included those results for comparison below. The samples from each cell line contained 7 groups at 3 time points (Fig 2a), in triplicate for every group. In the multidimensional scaling (MDS) analysis, each cell line clustered separately, and within the cell line clusters, untreated cells clustered apart from treated cells (S1 Fig).

The mapping rates for all samples were around 90% (S2 Table), which indicates high sequencing quality. Significantly differentially expressed (DE) genes were identified by comparing CKI treated cells to untreated cells in both cell lines (S3 Table sheet 1-4). DE genes from each individual cell line were compared to select the shared DE genes. This analysis

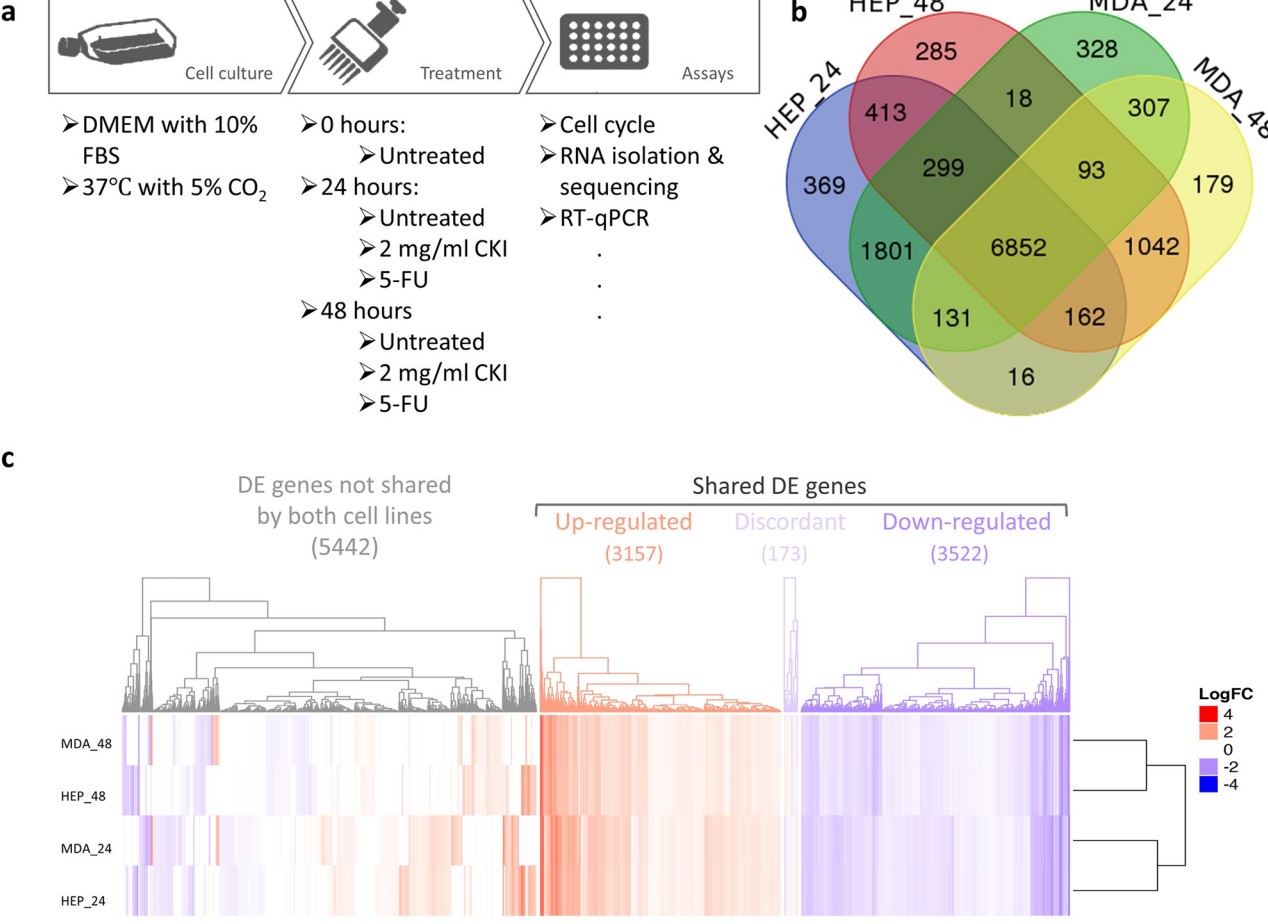

**Fig 2. DE genes shared in both cell lines at both time points.** a: Work flow diagram showing experimental design and sample collection. b: Venn diagram showing the number of shared DE genes between Hep G2 and MDA-MB-231. c: Heatmap presenting the overall gene expression pattern in both cell lines treated with CKI. Heatmap is split into four parts based on gene content and expression pattern: 5442 differentially regulated genes with expression not shared between the two cell lines, 3157 upregulated genes shared between both cell lines, 3522 down-regulated genes shared between both cell lines, and 173 discordantly regulated genes with differential expression shared between both cell lines.

generated thousands of DE genes (S3 Table sheet 5) between Hep G2 and MDA-MB-231 cell lines.

We identified a set of 6852 shared DE genes by identifying common DE genes between CKI-treated groups and untreated groups from both Hep G2 and MDA-MB-231 cell lines, at 24 hours and 48 hours (Fig 2b). These shared genes might predict a common molecular signature for CKI's activity. However, there were still a large number of DE genes that were not shared by both cell lines, as seen in the heatmap in Fig 2c. The expression of the shared gene set in both Hep G2 and MDA-MB-231 is highly consistent. Interestingly, this consistency is with respect to treatment time, rather than with respect to cell line.

## RT-qPCR validation and dose response of gene expression to CKI

Based on our previous results [24] and analysis above, we selected the four most significant DE genes expressed in G1-S phase of the cell cycle (TP53 and CCND1 for expression level validation and E2F2 and PCNA for low dose response test), as well as the proliferation and differentiation relevant ras subfamily encoding gene (RAP1GAP1) for low dose response test. We also

selected a prominently expressed gene (CYP1A1) for validation because of its sensitivity to CKI treatment. CYP1A1, TP53 and CCND1 expression changes were validated with RT-qPCR with all three genes showing similar patterns of expression in the transcriptome data and RT-qPCR (Fig 3a).

Because low dose treatment with CKI did not cause significant gross phenotypic effects in either cell line, we decided to use gene expression as a more sensitive measure of phenotype to look at the effect of lower doses of CKI. We used 0.125 mg/ml, 0.25 mg/ml, 0.5 mg/ml and 1 mg/ml concentrations to look for dose dependency of gene expression. Our results showed an obvious dose-dependent expression trend (Fig 3b) in both cell lines. Based on the recommended daily intravenous drip dosage of CKI (15-30 ml per day for adult) [13], a 0.125 mg/ml concentration of CKI is approximately equivalent to that experienced by a 50-60 kg cancer patient over 24 hours making our results potentially clinically relevant.

## Functional enrichment analysis

To identify candidate mechanisms of action of CKI, we carried out functional enrichment analysis. We used ConsensusPathDB [29] and Clusterprofiler [33] along with GO and KEGG pathways for over-representation analysis, along with disease ontology (DO) [35] enrichment.

GO over-representation was determined based on Biological Process at level 3 and q value <0.01. The results for both cell lines at both time points were summarized and visualized based on semantic analysis of terms in Fig 4a. From this result, it was obvious that there were a large proportion of enriched GO terms relating to cell cycle, such as "cell cycle checkpoint" and "negative/positive regulation of cell cycle process" which were prominently featured for all data sets (S2 Fig, S4 Table sheet 1-4).

We then used KEGG pathways to determine the specific pathways altered by CKI in cancer. The most regulated over-representative KEGG pathways are summarized according to KEGG Orthology (KO) (Fig 4b). Cell cycle related pathways such as "cell cycle", "DNA replication", and "apoptosis" were also consistently seen in the KEGG enrichment results (S4 Table sheet 5-8) at both 24 and 48 hours. Moreover, in addition to the cell cycle relevant pathways, some cancer-related pathways were also observed, such as "prostate cancer" and "chronic myeloid leukaemia", and a large number of DE genes (283) from the two cell lines were relevant in "pathways in cancer".

Because the KEGG enrichment revealed many pathways relating to diseases, most of which were cancers, we decided to explore the enrichment of DE genes with respect to DO terms (Fig 4c). In the DO list (S4 Table sheet 9-12), all top ranked terms are cancers. Interestingly, most listed cancer types are from the lower abdomen, for example, "ovarian cancer", "urinary bladder cancer "and "prostate cancer" etc. occurring in genitourinary organs (S4 Table sheet 9-12). For both KEGG pathway and DO enrichment, the effects of CKI on both cell lines were similar.

In addition to cell line specific functional enrichment of DE genes, we also analyzed the over-represented GO terms for shared DE genes (Fig 5a). The most significant clusters were highly relevant to metabolic processes, such as "cellular macromolecule metabolic process", as well as the corresponding positive/negative regulatory biological process (S4 Table sheet 13). Moreover, various signaling pathways, though not forming a large cluster, were also significant, for example, "regulation of signal transduction" and "intracellular receptor signaling pathway". Finally, some "cell cycle" related terms constituted relatively large sub-clusters, including "cell division" and "mitotic cell cycle process". The enriched GO analysis was consistent with the cell line specific enriched results, and with our previous analysis of MCF-7 cells

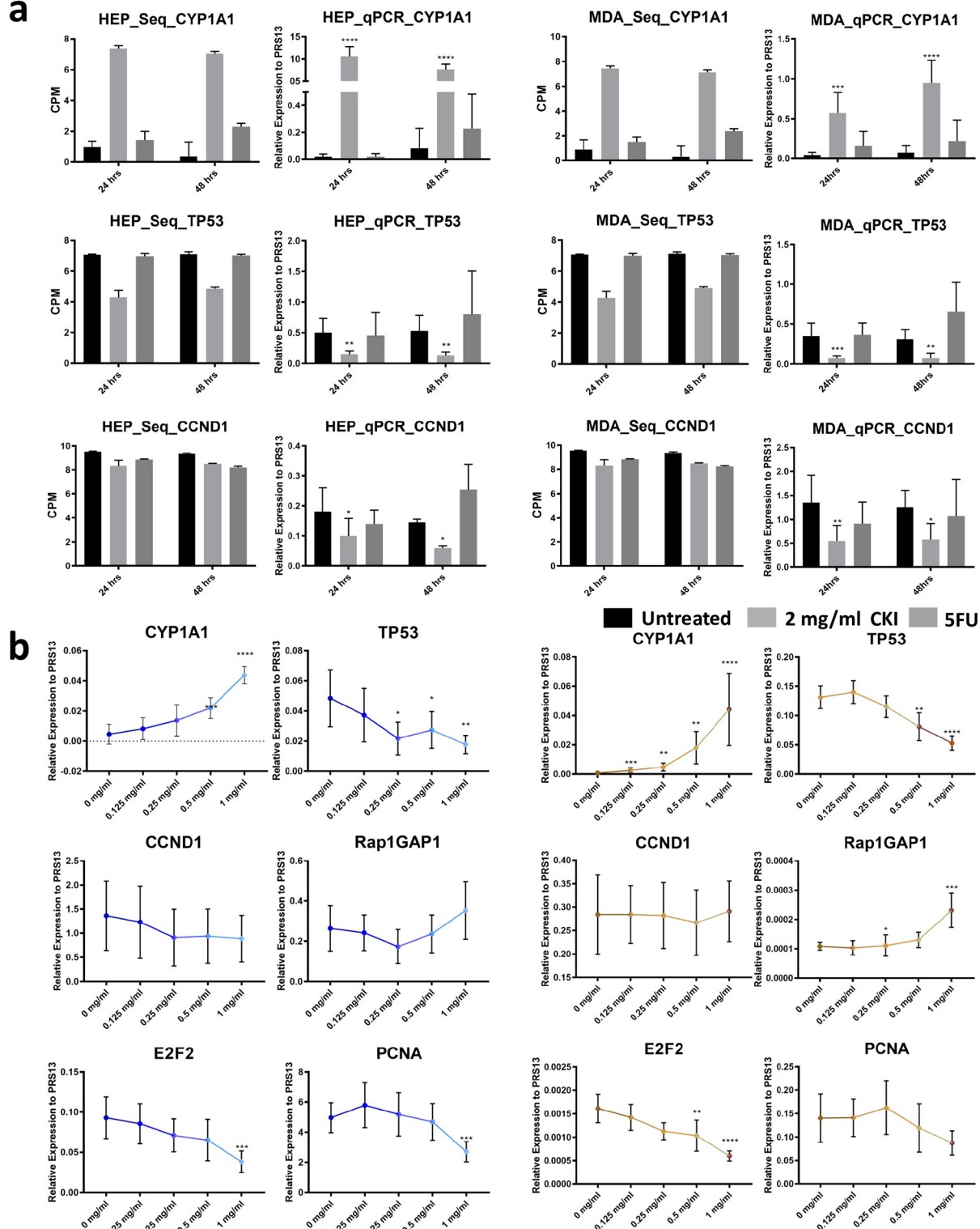

**Fig 3. Validation of gene expression and effects of low dose CKI using RT-qPCR.** a: Comparison of DE genes between RNA-seq results (left) and RT-qPCR validation (right) for each cell line at 2 time-points. Three DE genes (CYP1A1, TP53 and CCND1) were chosen for validation. Gene expression was generally consistent between transcriptome data and qPCR data. b: Dose response of CKI using a subset of genes with conserved expression in Hep G2 (left), and MDA-MB-231 (right) from 0 mg/ml to 1 mg/ml of total alkaloids. Six genes (CYP1A1, TP53, CCND1, Rap2GAP1, E2F2 and PCNA) were selected based on their relevance to important pathways perturbed by CKI. RT-qPCR

results are presented as expression relative to RPS13. Data are represented as mean ±95%CI (n>3). A t-test was used to compare CKI doses with "untreated" (*p<0.05, **p<0.01, ***p<0.001, ****p<0.0001).

[24]. It is worth noting that for "cell cycle" related terms, most of the participating genes were down-regulated by CKI.

Similar results were observed from KEGG analysis (Fig 5b, and S4 Table sheet 14) of shared genes. Various pathways related to cancer formed a large cluster. Pathways such as "DNA replication", "Ribosome" and "cell cycle" were mostly down-regulated, while up-regulated pathways included "inositol phosphate metabolism" and "protein processing in endoplasmic reticulum".

We also carried out an over-representation analysis of DO terms (Fig 5c) for all shared DE genes. The analysis results were consistent with the single cell line DO term analysis identifying mostly cancer-related terms; in particular genitourinary or breast cancer terms. While this was also partially similar to the KEGG results for shared DE genes, there were some differences in the KEGG results for disease pathways compared to the DO results, such as "bacterial invasion of epithelial cells", "Fanconi anemia pathway" and "AGE-RAGE pathway in diabetic complications".

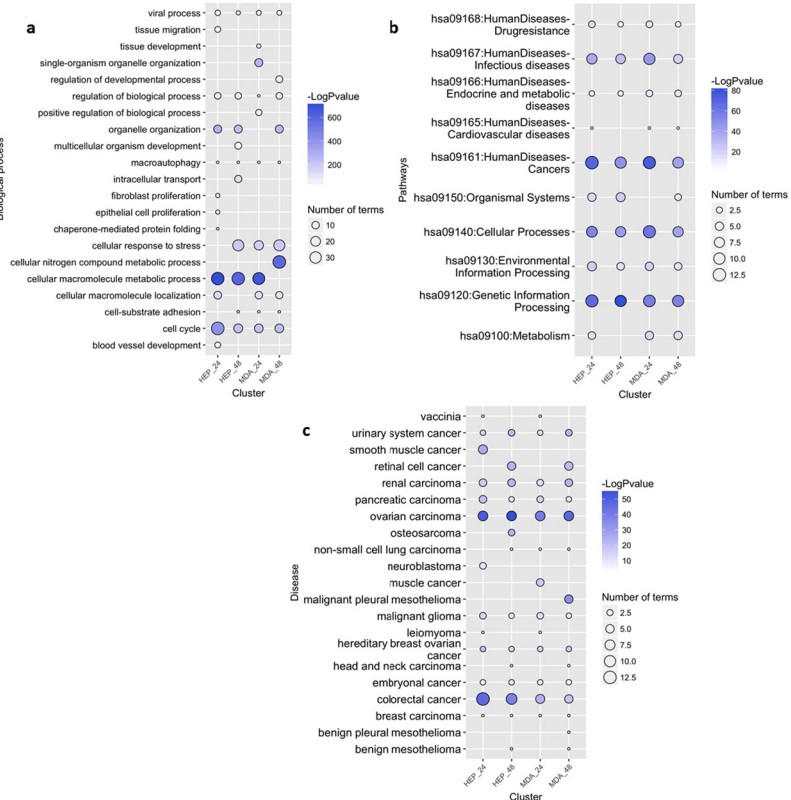

**Fig 4. Functional annotation of DE genes for each cell line as a result of CKI treatment.** Summary of over-represented a: GO terms for Biological Process, b: KEGG pathways and c: DO terms for DE genes as a result of CKI treatment in each cell line at two time points. For GO semantic and enrichment analysis, Lin's algorithm was applied to cluster and summarize similar functions based on GO terms found in every treatment. Similarly, by back-tracing the upstream categories in the KEGG Ontology, we were able to obtain a more generalized summary of KEGG pathways for each treatment. The size of each bubble represents the number of GO terms/pathways, and the colour shows the statistical significance of the relevant function or pathways. The DO summary for each treatment was determined by back-tracing to parent terms.

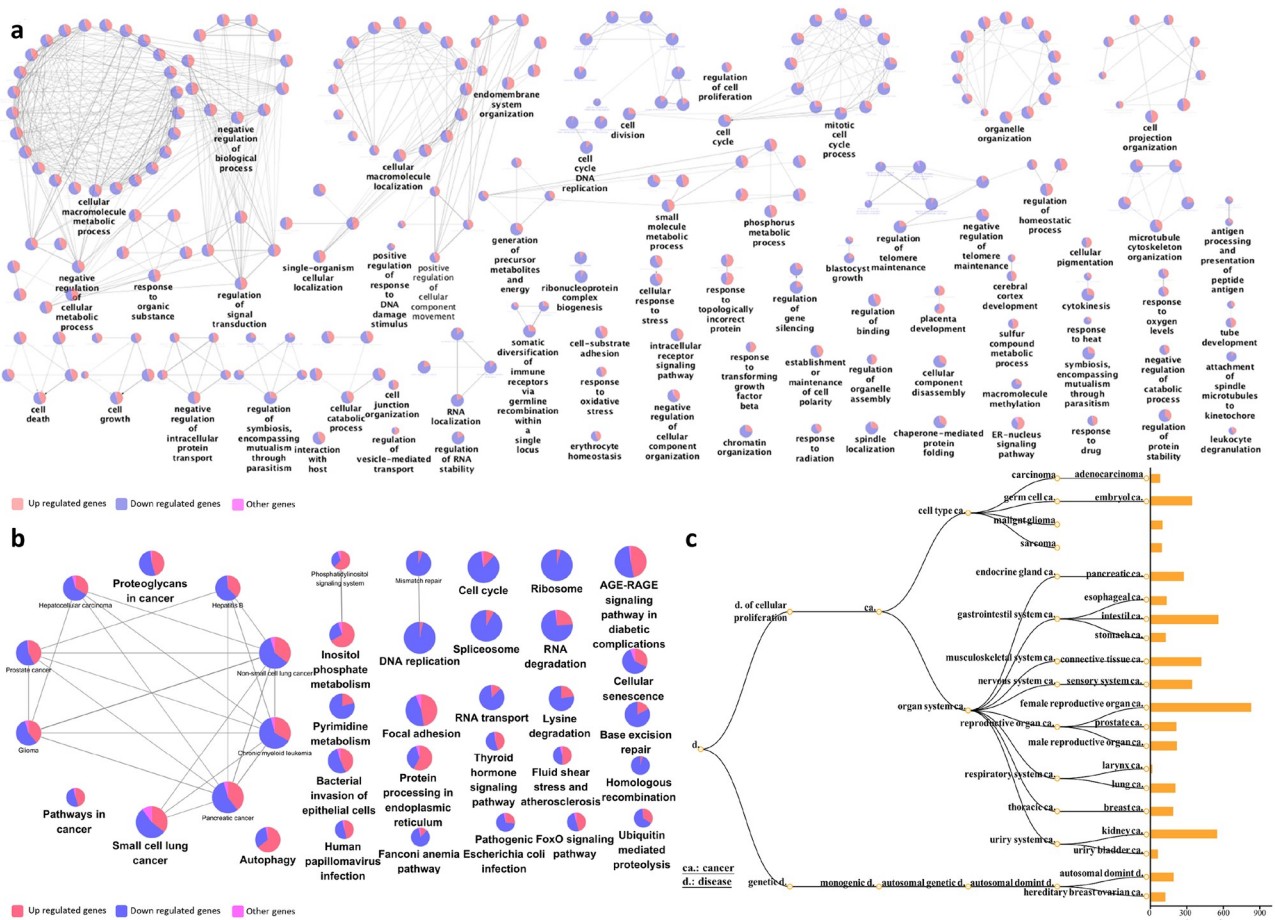

**Fig 5. Functional annotation of DE genes with shared expression in both cell lines as a result of CKI treatment.** Over-representation analysis was performed to determine a: GO terms for Biological Process, b: KEGG pathways, and c: DO terms for DE genes shared in both cell lines. In nodes for both GO terms and KEGG pathways, node size is proportional to the statistical significance of over-representation. For DO terms, all the enriched terms are statistically significant (p<1 × 10$^{-5}$) in each category, and the bar length represents the number of expressed genes that map to the term.

Specific to the therapeutic potential of CKI for cancer treatment, we applied our data set mapping to KEGG cancer pathways: pathways in cancer—homo sapiens (S3 Fig). The R package Pathview [31] was used to integrate log fold change values of all the gene expression patterns into these target pathways. Within the 21 pathways in cancer, the "cell cycle" still featured prominently (Fig 6a). The expression of almost every gene in the cell cycle pathway was affected by CKI, with most of them suppressed. We did not observe this kind of overall pathway suppression in any of the other pathways. We have displayed the summaries for the remaining 20 pathways in the heatmap in Fig 6b. Although all the pathways were all perturbed by CKI, they include both over and under expressed genes in roughly equal proportions.

Collectively, these results suggest a direct anti-cancer effect of CKI and implicate specific candidate mechanisms of action based on the perturbed molecular networks. The most obvious example is the cell cycle, where the G1-S phase is significantly altered, resulting in the induction of apoptosis. The downstream process triggered by CKI is the suppression of gene expression of cell cycle regulators, including TP53 and CCND1. The other perturbed cancer pathways provide additional candidate mechanisms of action for CKI. In the following section,

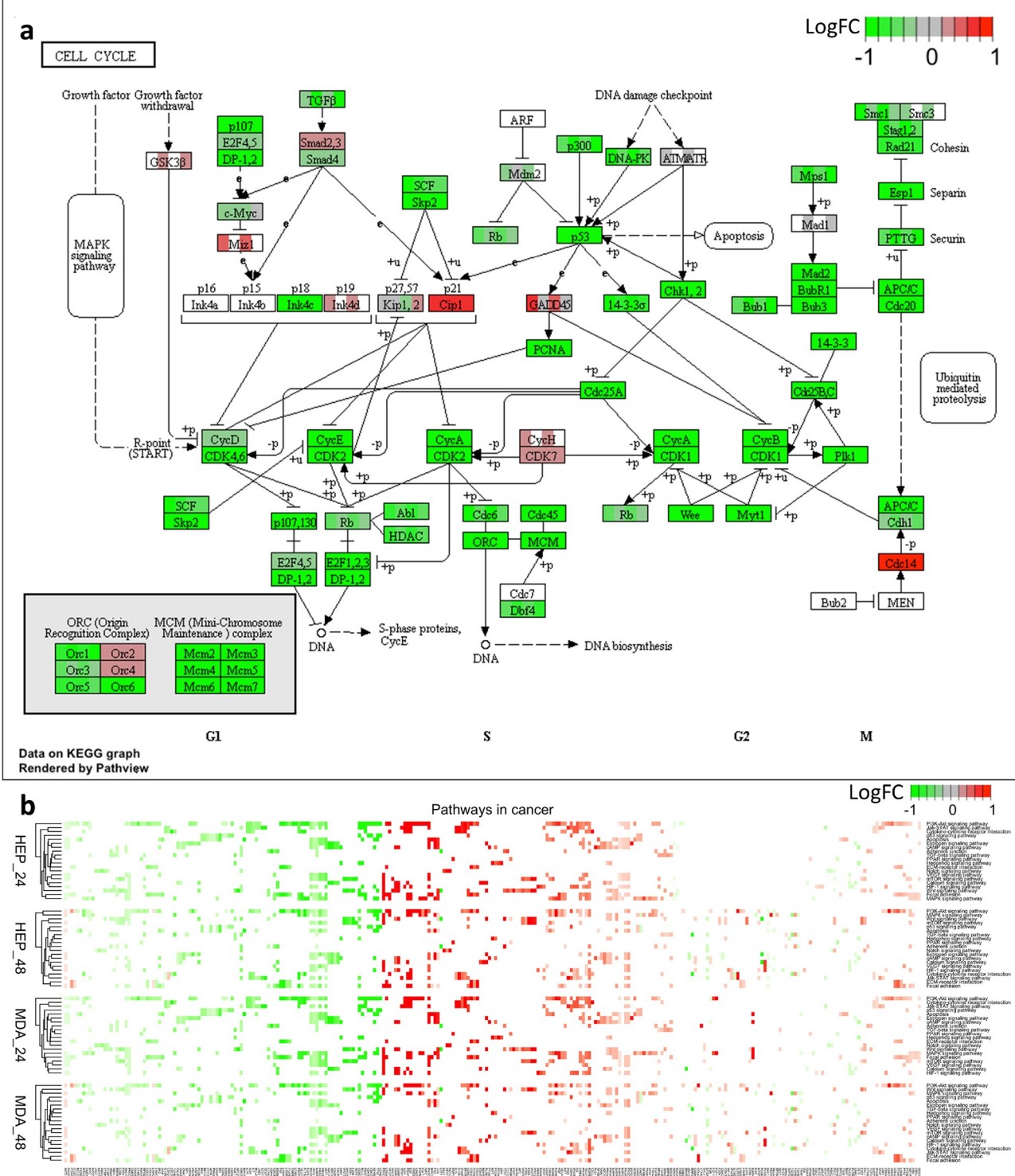

**Fig 6. Comparison of shared genes expression in specific pathways across two cell lines.** a: Cell cycle pathways, where each coloured box is separated into 4 parts, from left to right representing 24 hour CKI treated Hep G2, 48 hour CKI treated Hep G2, 24 hour CKI treated MDA-MB-231 as well as 48 hour CKI treated MDA-MB-231. b: Heatmap of pathways in cancer. The top two heatmaps summarise the effects of CKI on Hep G2 cells for two time-points, and the bottom two heatmaps show the effects of CKI on MDA-MB-231 cells. In addition to the cell cycle pathway, there were 21 associated pathways in cancer that were perturbed by CKI. The effects of CKI on both cell lines were similar, with changes in TARGET database genes indicated by arrows. Compared to other pathways in cancer, the effects of CKI on the cell cycle pathway showed overall down-regulation.

we integrate these results with previous results reported in the literature to refine the core set of genes and pathways perturbed by CKI.

## Discussion

Although Hep G2 (liver cancer—mesodermal tissue origin) and MDA-MB-231 (mammary epithelial adenocarcinoma—ectodermal tissue origin) are different cancer types, they shared a large number of CKI DE genes with similar expression profiles, presumably, these shared genes include CKI response genes that are essential to the apoptotic response triggered by CKI. However, the number of shared CKI DE genes is too high to allow straight forward identification of genes critical to the CKI response. We, therefore, decided to combine these data with previously reported CKI DE genes from MCF-7 cells [24] in order to reduce the number of core CKI response genes. The intersection of MCF-7 CKI DE genes with the shared CKI DE genes yielded 363 core CKI DE genes (S4 Fig).

Among the 363 core CKI DE genes, cytochrome P450 family 1 subfamily A member 1 (CYP1A1) gene is the most over-expressed. This gene is consistently up-regulated by CKI in all three cell lines and showed a significant dose response. In liver cancer cells, over-expression of CYP1A1 induced by plant natural products has been associated with Aryl-hydrocarbon Receptor transformation [36, 37]. Furthermore, as a steroid-metabolizing enzyme, CYP1A1 is part of cancer metabolic processes relevant to steroid hormone responsive tumors, such as breast cancer, ovarian cancer, and prostate cancer [38–41]. Therefore, CYP1A1 may be of particular interest in understanding the mechanism of action of CKI on cancer cells.

Comparison of the 363 core genes to the 135 Tumor Alterations Relevant for Genomics-driven Therapy (TARGET) genes (version 3) from The Broad Institute (https://www.broadinstitute.org/cancer/cga/target) identified 7 DE genes that were shared across the three cell lines and two time points (Fig 7a). Of these seven genes, six (TP53, CCND1, MYD88 (Myeloid differentiation primary response gene 88), EWSR1, TMPRSS2 and IDH1 (isocitrate dehydrogenase 1) were similarly regulated (either always over-expressed or under-expressed), while CCND3 was over-expressed in all three cell lines at both time points except at 48 hours in MCF-7 cells, where it was under-expressed.

The TP53 gene encodes a tumor suppressor protein, that can induce apoptosis [42]. However, in all cell lines, TP53 was down-regulated, and all cell lines showed increased apoptosis. This suggests that CKI induced apoptosis was not TP53-dependent. Support for this comes from the fact that transcripts for PCNA (proliferating cell nuclear antigen), and a group of transcription factors: MCM (mini-chromosome maintenance) complex and the E2F family are down-regulated. The E2F transcription factors regulate the cell cycle and TP53-dependent and -independent apoptosis [43–46]. In addition, other core genes present in the TARGET database have also been shown to induce apoptosis. For example, inhibition of MYD88 induces apoptosis in both triple negative breast cancer and bladder cancer [47, 48]. The increased expression of IDH1 may be important, as IDH1 is frequently mutated in cancers [33] and when mutated, it causes loss of $\alpha$-ketoglutarate production and may be important for the Warburg effect. TMPRSS2 (transmembrane protease, serine 2) has also been shown to regulate apoptosis in cancer [49]. Therefore, CKI may induce apoptosis through a variety of means.

The GO (Fig 7b) and KEGG (Fig 7c) over-representation analysis of the 363 core genes yielded enrichment for cell cycle and cancer pathways. In the GO enriched genes, cell cycle and related pathways accounted for the majority of functional sub-clusters. In the KEGG enriched pathways, cell cycle and cancer pathways predominated in a single cluster. Most of the core genes in GO and KEGG clusters were down-regulated by CKI. In addition to the cell

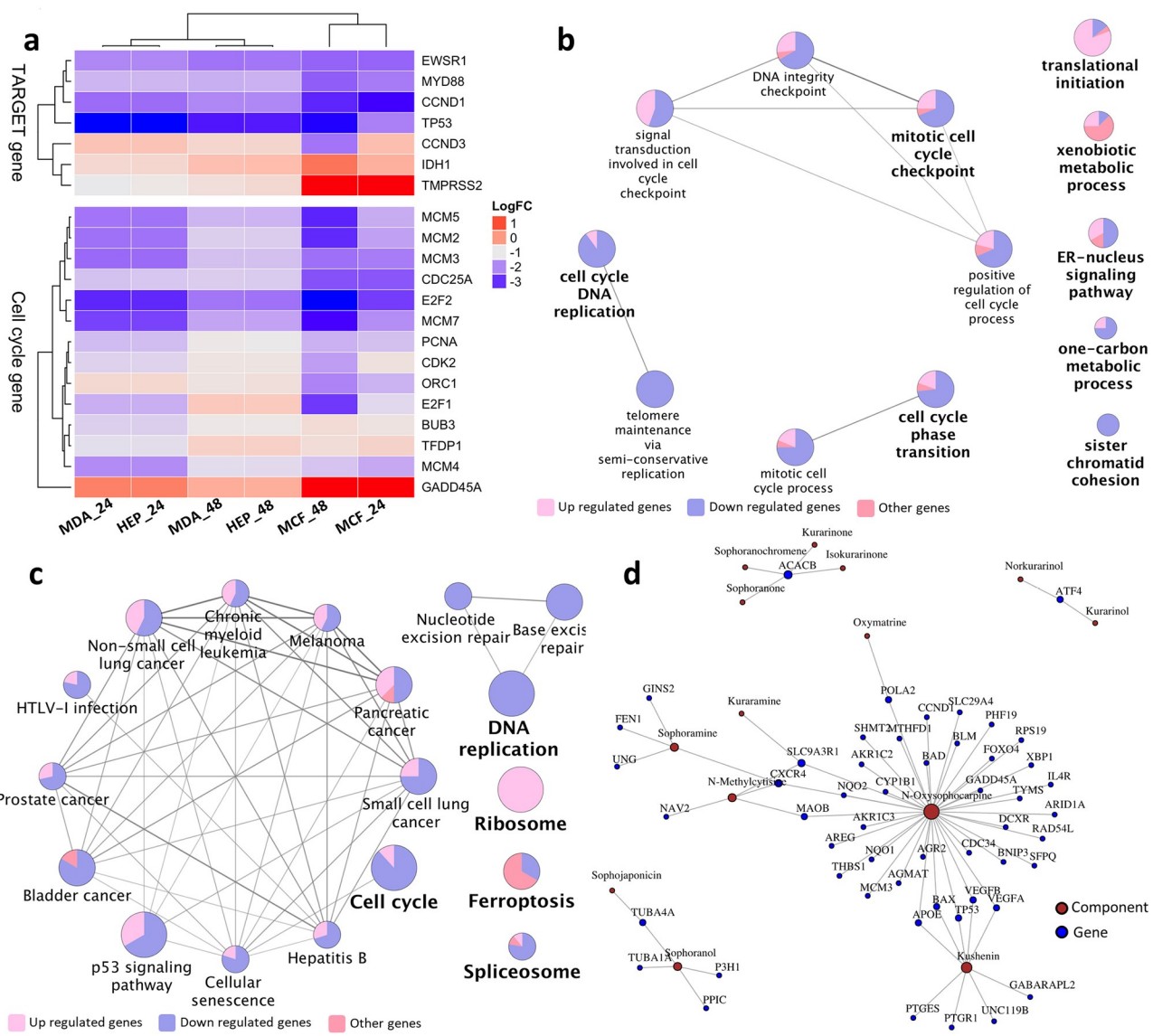

**Fig 7. Analysis of CKI regulated core genes from this report combined with previous available data.** a: Fold changes of TARGET and cell cycle regulatory gene expression in MDA-MB-231, Hep G2 and MCF-7 [24] cell lines 24 and 48 hours after CKI treatment. Only seven TARGET genes are affected by CKI in all three cell lines. Most of the 14 cell cycle regulatory genes differentially expressed in all three cell lines are down-regulated. b: GO term enrichment analysis of 363 core genes from MDA-MB-231, Hep G2 and MCF-7 cell lines. c: KEGG pathway enrichment of 363 core genes from MDA-MB-231, Hep G2 and MCF-7 cell lines. d: Some individual compounds present in CKI linked to genes they regulate that are also found in this report and our previous study [24]. Node size is proportional to the number of related components/genes.

cycle, CKI treatment also caused enrichment for terms or pathways related to cancer progression, such as "focal adhesion" and "blood vessel development". (S4 Table sheet 5-8). These developmental processes contribute to tumorigenesis and metastasis [50, 51]. It is tempting to speculate that CKI may alter these functions *in vivo*, possibly altering angiogenesis which is critical for tumor progression [52]. In addition, there were metabolic pathways and terms that were also identified as perturbed by CKI. Effects on many targets/pathways are expected features of TCM drugs which likely hit multiple targets [53]. In a previous study, we have

reported in depth analysis of several pathways and validated the expression of associated key proteins [23].

We have examined the effect of a complex mixture of plant natural products (CKI) on different cancer cell lines and have identified specific, consistent effects on gene expression resulting from this mixture. However, the complexity of CKI makes it difficult to determine the mechanism of action of individual components, and often testing of individual components has resulted in either no effect or contradictory results in the research literature. In spite of this complexity, it is possible to map our results on to a pre-existing corpus of work that links individual natural compounds to changes in gene expression. We have used BATMAN [32], an online TCM database of curated links between compounds and gene expression. Due to the limited numbers of well researched TCM compounds, based on this resource, we have identified 8 components of CKI that have been linked to the regulation of 52 of our core genes (Fig 7d). We can see from the network diagram in Fig 7d that one to one, one to many and many to many relationships exist between CKI components and genes which is consistent with previous studies [7, 14, 54]. While the compounds that are not characterised/listed may also have this potential to regulate different numbers of genes, as more information becomes available for individual components, we will be able to construct a more comprehensive model of CKI mechanism based on network analysis.

## Conclusion

Our systematic analysis of gene expression changes in cancer cells caused by a complex herbal extract used in TCM has proven to be effective at identifying candidate molecular pathways. CKI has consistent and specific effects on gene expression across multiple cancer cell lines and it also consistently induces apoptosis *in vitro*. These effects show that CKI can suppress the expression of cell cycle regulatory genes and other well-characterized cancer-related genes and pathways. Validation of a subset of DE genes at lower doses of CKI has shown a dose-response relationship which suggests that CKI may have similar effects *in vivo* at clinically relevant concentrations. Our results provide a molecular basis for further investigation of the mechanism of action of CKI.

## Supporting information

**S1 Fig. MDS plot of the DE gene distribution of two cell lines under different conditions.** (TIF)

**S2 Fig. GO semantic similarity analysis of each data set.** (A-D) Each small square represents a GO biological process function at level 3. Size of the squares positively correlates with the statistical significance of related biological process. Different colours distinguish biological process clusters that are described by the top shaded functional representatives. (PDF)

**S3 Fig. Distribution of DE genes of two cell lines in pathways in cancer.** In the cell cycle pathway, each coloured box is separated into 4 parts, from left to right representing 24h CKI treated Hep G2, 48h CKI treated Hep G2, 24h CKI treated MDA-MB-231 and 48h CKI treated MDA-MB-231. (TIF)

**S4 Fig. Heatmap of core genes with expression altered by CKI in three cell lines.** Heatmap showing the expression fold changes of core genes in three cell lines at two time points. All the core genes can be separated into the following 3 clusters: genes up regulated in all three cell lines, genes down regulated in all three cell lines and DE genes that are uncorrelated in terms

of expression change across the three cell lines.
(TIF)

**S1 Table. RT-qPCR target genes and their primer sequences.**
(XLSX)

**S2 Table. Mapping rate of each cell line.**
(XLSX)

**S3 Table. List of DE genes in each cell line at each time point.**
(XLSX)

**S4 Table. Summary of functional analysis of separate datasets and shared datasets.** Sheet 1-4: GO enrichment of each cell line at two time points. Selection standard: cut off $p$ value<0.01, cut off $q$ value<0.01. Sheet 5-8: KEGG enrichment of each cell line at two time points. Selection standard: cut off $p$ value<0.01, cut off $q$ value<0.01. Sheet 9-12: DO enrichment of each cell line at two time points. Selection standard: cut off $p$ value<0.01, cut off $q$ value<0.01. Sheet 13: GO enrichment of shared genes by both cell lines. Selection standard: cut off $p$ value<0.01. Sheet 14: KEGG enrichment of shared genes by both cell lines. Selection standard: cut off $p$ value<0.01.
(XLSX)

**S1 File. Quality control report from Zhendong Pharmaceutical Co. Ltd for the batch of CKI used in these experiments.** Both original Chinese document and English translation included.
(PDF)

## Acknowledgments

DLA would like to thank Rory A. for inspiration and support.

## Author Contributions

**Conceptualization:** Jian Cui, Zhipeng Qu, Yuka Harata-Lee, David L. Adelson.

**Data curation:** Jian Cui.

**Formal analysis:** Jian Cui, Zhipeng Qu, Yuka Harata-Lee.

**Funding acquisition:** Wei Wang, David L. Adelson.

**Investigation:** Jian Cui, Yuka Harata-Lee.

**Methodology:** Jian Cui, Zhipeng Qu, Yuka Harata-Lee.

**Project administration:** Yuka Harata-Lee, David L. Adelson.

**Resources:** Zhipeng Qu, Wei Wang.

**Supervision:** Zhipeng Qu, Yuka Harata-Lee, R. Daniel Kortschak, David L. Adelson.

**Validation:** Hanyuan Shen, Thazin Nwe Aung, Wei Wang.

**Writing – original draft:** Jian Cui, Zhipeng Qu, Yuka Harata-Lee, David L. Adelson.

**Writing – review & editing:** Jian Cui, Zhipeng Qu, Yuka Harata-Lee, R. Daniel Kortschak, David L. Adelson.

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
