## [Decision Letter · Decision Letter 0]

28 May 2020

PONE-D-20-11981

The effect of compound kushen injection on cancer cells: integrated identification of candidate molecular mechanisms

PLOS ONE

Dear Dr. Adelson,

Thank you for submitting your manuscript to PLOS ONE. After careful consideration, we feel that it has merit but does not fully meet PLOS ONE’s publication criteria as it currently stands. Therefore, we invite you to submit a revised version of the manuscript that addresses the points raised during the review process.

Both reviewers have raised some issues but those of Reviewer 2 are particularly of concern and need to be addressed if the authors plan to submit a revised manuscript.

We look forward to receiving your revised manuscript.

Kind regards,

Salvatore V Pizzo

Academic Editor

PLOS ONE

2. Please provide additional information about the MDA-MB-231 and HepG2 cells used in this work, including any quality control testing procedures (authentication, characterisation, and mycoplasma testing). For more information, please see http://journals.plos.org/plosone/s/submission-guidelines#loc-cell-lines.

3. We understand that you obtained Compound Kushen Injection (CKI)  from Zhendong Pharmaceutical Co. Ltd for this study. For purposes of reporting, we request that you provide additional details as to the source of this material. Please provide any quality assessments and chemical assessments that were supplied when you obtained the product.

4.  To comply with PLOS ONE submission guidelines, in your Methods section, please provide additional information regarding your statistical analyses. For more information on PLOS ONE's expectations for statistical reporting, please see https://journals.plos.org/plosone/s/submission-guidelines.#loc-statistical-reporting.

"We wish to draw the attention of the Editor to the following facts which may be

considered as potential conflicts of interest and to significant financial contributions to

this work. While a generous donation was used to set up the Zhendong Centre by

Shanxi Zhendong Pharmaceutical Co Ltd, they did not determine the research direction

for this work or influence the analysis of the data. JC: no competing interests, ZQ: no

competing interests, YHL: no competing interests, HS: no competing interests, TNA: no

competing interests, WW: is an employee of Zhendong Pharma seconded to Zhendong

Centre to learn bioinformatics methods, RDK: no competing interests, DLA: Director of

the Zhendong Centre which was set up with a generous donation from the Zhendong

Pharmaceutical Co Ltd. Zhendong Pharmaceutical has had no control over these

experiments, their design or analysis and have not exercised any editorial control over

the manuscript."

We note that you received funding from a commercial source: Shanxi Zhendong Pharmaceutical Co Ltd.,

Reviewers' comments:

Reviewer's Responses to Questions

**Comments to the Author**

1. Is the manuscript technically sound, and do the data support the conclusions?

Reviewer #1: Yes

Reviewer #2: No

2. Has the statistical analysis been performed appropriately and rigorously? 

Reviewer #1: Yes

Reviewer #2: Yes

3. Have the authors made all data underlying the findings in their manuscript fully available?

Reviewer #1: Yes

Reviewer #2: Yes

4. Is the manuscript presented in an intelligible fashion and written in standard English?

Reviewer #1: Yes

Reviewer #2: Yes

5. Review Comments to the Author

Reviewer #1: This is a good conduced experiment that made a comprehensive exploration of the mechanism of compound kushen injection in two cancer cell lines. Overall, the methodology used is very adequate. However, some points should be taken in consideration to improve the manuscript.

(1) There are some mistakes in English and grammatical errors. The authors should made a critical review in language;

(2) Both MCF-7 and MDA-MB-231 are breast cancer cell lines but they have the different endocrine genotype. Is it reasonable to integrate the data of MCF-7 and MDA-MB-231 cell lines together?

(3) Maybe the authors can prove theirs hypothesis in vivo the next step.

Reviewer #2: CKI is so broadly utilized that understanding it's anticancer function is critical to the world's population, however, due to concerns with the experimental design of this study the anticancer activity of CKI will not be revealed here. 1.The comparison of gene expression between two cell lines that originated from very different tissues (breast vs. liver) without including immortalized (BUT NOT TRANSFORMED) tissue type cell line matches are likely to highlight a subset of DNA damage and repair genes common across all tissues and species. It is unclear that CKI drives apoptosis in vivo. However att doses of 1 mg/ml and 2 mg/ml in an in vitro setting where aryl hydrocarbon receptors could be activated cell death will be induced. This is seen in manuscript Figure 1. Therefore the experimental design biases the outcome of the study toward.

2. PRS13 is regulated by cell cycle arrest Fannie W. Chen & Yiannis A. Ioannou (1999) Ribosomal Proteins in Cell Proliferation and Apoptosis, International Reviews of Immunology, 18:5-6, 429-448, DOI: 10.3109/08830189909088492, yet it's expression is the reference gene for this study - See Figure 3.

3. Figure 4 does not address the question being asked. Some genes that are altered by CKI treatment of HEPG2 and MDA MB 231 cells correlate with other cancer types. What does that suggest about the anticancer mechanism of CKI?

4. It is unclear that the mutant form of TP 53 expressed by MDA MB 231 and the wild type form of TP 53 expressed by HEPG2 should be equally regulated by CKI. Figure 3 shows temporal regulation of TP53 in both cell types that is reversed at later time points. If real, this could be exciting but the how and why of this is not explored nor is it convincing due to the differences (or lack there of) the two methods to quantify the results.

Thus while there is a great deal of reasonable bioinformatics data generated in this study, due to the experimental design, it does not reveal a mechanism for CKI anti-cancer activity in an unbiased way.

6. PLOS authors have the option to publish the peer review history of their article (what does this mean?). If published, this will include your full peer review and any attached files.

Reviewer #1: No

Reviewer #2: No

---

## [Author Response · Author response to Decision Letter 0]

23 Jun 2020

Reviewer #1: This is a good conduced experiment that made a comprehensive exploration of the mechanism of compound kushen injection in two cancer cell lines. Overall, the methodology used is very adequate. However, some points should be taken in consideration to improve the manuscript.

(1) There are some mistakes in English and grammatical errors. The authors should made a critical review in language;

RESPONSE. We have made a critical review of the manuscript and corrected mistakes or grammatical errors. 

(2) Both MCF-7 and MDA-MB-231 are breast cancer cell lines but they have the different endocrine genotype. Is it reasonable to integrate the data of MCF-7 and MDA-MB-231 cell lines together?

RESPONSE. Our purpose in comparing these three cell lines was to reveal the shared anticancer mechanisms of CKI in different cancer types. A previously conducted assessment of CKI effects on an array of human cancer cell lines (NCI60 panel) indicated that these three cell lines were amongst the most sensitive to CKI treatment. Therefore, we elected evaluate the molecular mechanisms of CKI effects within the same cancer type (but with different genotypes) or across different cancer types. As a result, our study has shown that the anticancer mechanisms of CKI are centered on arresting the cell cycle and include other pathways that trigger cancer apoptosis. This provides a common set of molecular candidates across varied endocrine genotypes of cancer cell lines and varied cancer tissue types. 

(3) Maybe the authors can prove theirs hypothesis in vivo the next step.

RESPONSE. We agree. Indeed, our next research strategy is to carry out studies in vivo, such as in drosophila or mice, and we have already carried out preliminary work in this regard. This will be a story for another manuscript.

Reviewer #2: CKI is so broadly utilized that understanding it's anticancer function is critical to the world's population, however, due to concerns with the experimental design of this study the anticancer activity of CKI will not be revealed here. 

 1. The comparison of gene expression between two cell lines that originated from very different tissues (breast vs. liver) without including immortalized (BUT NOT TRANSFORMED) tissue type cell line matches are likely to highlight a subset of DNA damage and repair genes common across all tissues and species. It is unclear that CKI drives apoptosis in vivo. However att doses of 1 mg/ml and 2 mg/ml in an in vitro setting where aryl hydrocarbon receptors could be activated cell death will be induced. This is seen in manuscript Figure 1. Therefore the experimental design biases the outcome of the study toward.

RESPONSE. We do not believe the inclusion of immortalised cell lines as controls would provide a good comparison as they are genetically modified to behave more like cancer cells than primary cell lines. 

First, we would like to clarify that we are not directly comparing transcriptomes between two cell lines that originated from two different tissues (MDA-MB-231 from breast vs. Hep G2 from liver). In our study, the transcriptome of CKI-treated cells was compared with corresponding untreated cells to generate a list of DE (differentially expressed) genes and pathways that are significantly affect by CKI treatment. This was done separately for each cell line. Subsequently, we compared the DE genes/pathway list of three cell lines and to look for common DE genes and pathways that CKI affects in all three cell/tissue types. We believe that these common DE genes and altered pathways across three cancer cell lines is the real reflection of molecular changes caused by CKI at gene level in different cancer types. Therefore, the result of this study showed in an unbiased fashion that CKI can affect cell cycle arrest of not only one type of cancer but various types of cancers. We do not believe the inclusion of immortalised cell lines as controls would provide a good comparison as they are genetically modified to be immortal, like cancer cells. See below for more on this in the response to point 4. 

The possibility of aryl hydrocarbon receptors being activated by CKI does not invalidate our approach. It may be that this response is part of the effect CKI has on cancer cells, but that it cooperates with other mechanisms that are also pro-apoptotic. At this point we cannot answer this question, but it is beyond the scope of our present study, which aimed to identify conserved DE gene responses to CKI. 

 2. PRS13 is regulated by cell cycle arrest Fannie W. Chen & Yiannis A. Ioannou (1999) Ribosomal Proteins in Cell Proliferation and Apoptosis, International Reviews of Immunology, 18:5-6, 429-448, DOI: 10.3109/08830189909088492, yet it's expression is the reference gene for this study - See Figure 3.

RESPONSE. We selected RPS13 as the reference gene for our study based on the findings from Hendrik J. M. de Jonge et al., 2007, PLoS One and Francis Jacob et al., 2013, PLoS One. According to these reports, RPS13 is one of the most stably expressed genes in the most commonly used 50 cell lines and is the 2nd or 3rd most stably expressed gene across 25 cell lines, including cancer cell lines. No reference gene will be perfect for every condition, but RPS13 is a good choice in this instance. RPS13 is used by many investigators as a “housekeeping” gene for RT-qPCR in studies focusing on cancer cells. 

 3. Figure 4 does not address the question being asked. Some genes that are altered by CKI treatment of HEPG2 and MDA MB 231 cells correlate with other cancer types. What does that suggest about the anticancer mechanism of CKI?

RESPONSE.

Many cancer-associated genes play important roles in multiple types of cancers. As we clarified in Question 1, the gene set that we used for functional enrichment analysis in Figure 4 is common DE genes in both HepG2 and MDA-MB-231 cell lines due to CKI treatment, which should be mostly under the category of generalized/essential cancer-associated genes across multiple cancer types. The results suggest that CKI can alter the expression of many essential genes that are not cancer-type specific, but found in cancer cells. This is consistent with the clinical use of CKI for treating different types of cancers. 

 4. It is unclear that the mutant form of TP 53 expressed by MDA MB 231 and the wild type form of TP 53 expressed by HEPG2 should be equally regulated by CKI. Figure 3 shows temporal regulation of TP53 in both cell types that is reversed at later time points. If real, this could be exciting but the how and why of this is not explored nor is it convincing due to the differences (or lack there of) the two methods to quantify the results.

RESPONSE. 

Based on our results in Figure 3, TP53 was regulated by CKI in a similar fashion in both cell lines (statistically significant down-regulated by CKI at both time points), which indicates the expression change of TP53 altered by CKI is independent of its mutation status. With respect to the statement of “temporal regulation of TP53 in both cell types that is reversed at later time points”, we don’t see this reversed regulation in Figure 3. 

Thus while there is a great deal of reasonable bioinformatics data generated in this study, due to the experimental design, it does not reveal a mechanism for CKI anti-cancer activity in an unbiased way.

RESPONSE. 

We believe this approach is unbiased in that it looks at all genes in the same way. Their differential expression then identifies them as important in terms of CKI activity and mechanism. The inclusion of non-cancer cell lines would not necessarily make the study more “unbiased” as similar pathways might be affected in those cells. It would not be reasonable to rule out those pathways just because they are also identified by DE genes in non-cancer cells. We know that many cancer therapies affect cancer cells and non-cancer cells, but that does not mean they are not useful therapies.

---

## [Editor Report · Decision Letter 1]

8 Jul 2020

The effect of compound kushen injection on cancer cells: integrated identification of candidate molecular mechanisms

PONE-D-20-11981R1

Dear Dr. Adelson,

We’re pleased to inform you that your manuscript has been judged scientifically suitable for publication and will be formally accepted for publication once it meets all outstanding technical requirements.

Kind regards,

Salvatore V Pizzo

Academic Editor

PLOS ONE
---

## [Editor Report · Acceptance letter]

13 Jul 2020

PONE-D-20-11981R1 

The effect of compound kushen injection on cancer cells: integrated identification of candidate molecular mechanisms 

Dear Dr. Adelson:

I'm pleased to inform you that your manuscript has been deemed suitable for publication in PLOS ONE. Congratulations! Your manuscript is now with our production department. 

Kind regards, 

on behalf of

Dr. Salvatore V Pizzo 

Academic Editor

PLOS ONE